# Valorization of Grain and Oil By-Products with Special Focus on Hemicellulose Modification

**DOI:** 10.3390/polym16121750

**Published:** 2024-06-20

**Authors:** Xiaoxian Liu, Jin Xie, Nicolas Jacquet, Christophe Blecker

**Affiliations:** Gembloux Agro-Bio Tech, Unit of Food Science and Formulation, University of Liège, Avenue de la Faculté d’Agronomie 2B, B-5030 Gembloux, Belgium; xiaoxian.liu@doct.uliege.be (X.L.); jin.xie@student.uliege.be (J.X.)

**Keywords:** grain by-products, oil by-products, hemicellulose, arabinoxylan, xylan, modification, industrial application

## Abstract

Hemicellulose is one of the most important natural polysaccharides in nature. Hemicellulose from different sources varies in chemical composition and structure, which in turn affects the modification effects and industrial applications. Grain and oil by-products (GOBPs) are important raw materials for hemicellulose. This article reviews the modification methods of hemicellulose in GOBPs. The effects of chemical and physical modification methods on the properties of GOBP hemicellulose biomaterials are evaluated. The potential applications of modified GOBP hemicellulose are discussed, including its use in film production, hydrogel formation, three-dimensional (3D) printing materials, and adsorbents for environmental remediation. The limitations and future recommendations are also proposed to provide theoretical foundations and technical support for the efficient utilization of these by-products.

## 1. Introduction

The increasing scarcity of non-renewable resources has gradually become a stumbling block on the path to sustainable human development. Therefore, people are constantly searching for renewable and environmentally friendly materials to promote environmental sustainability. Grain and oil processing industries produce many by-products, mainly including bran, straw, and seed hulls. These by-products account for about one-third of the total output of grains and oils [1]. Compared to other biomass resources, grain and oil by-products (GOBPs) are naturally generated during the production of main products (such as flour), meaning the cost of using these by-products as raw materials is lower, as they are already produced without additional planting or harvesting costs. The effective conversion and recycling of by-product resources are of great significance to humanity [2]. However, due to the difficulties in storing and transporting these by-products, they are usually used as feed, fertilizer, or left directly in the fields. These methods have low economic benefits, and burning the residues to clear the fields leads to severe air pollution and resource waste, seriously impacting the ecosystems and human health [3]. To reduce environmental pollution and achieve sustainable development, it is necessary to utilize GOBPs to produce high-value products. Hemicellulose modification has become an important branch in the research of high-value utilization of GOBPs.

Hemicellulose, primarily composed of xylan, is a complex natural polysaccharide that comprises about 20%–35% of plant biomass [4]. It is a rich, biocompatible, and renewable material. Hemicellulose is widely used in the production of food, materials, pharmaceuticals, and chemicals [5]. However, the chemical and molecular structures of hemicellulose vary, as shown in Figure 1; the hemicellulose of grasses primarily consists of arabinoxylan and mixed-linkage glucan. Arabinoxylan has a backbone composed of (1→4)-*β*-d-xylopyranose units with side chains of arabinose attached through *α*-(1→2) and/or *α*-(1→3) linkages; mixed-linkage glucan is composed of glucose units connected by alternating (1→3) and (1→4) linkages (Figure 1A). In hardwoods, the main component of hemicellulose is O-acetyl-4-O-methylglucuronoxylan, with a backbone of (1→4)-*β*-d-xylopyranose units and side chains of 4-O-methyl-*α*-d-glucuronic acid residues (Figure 1B). In softwoods, the predominant hemicellulose is O-acetyl-galactoglucomannan, which consists of a backbone of (1→4)-linked *β*-d-mannopyranose and *β*-d-glucopyranose units, with acetyl groups substituting at the C-2 and C-3 positions (Figure 1C). These hemicelluloses exhibit amorphous structures, high degrees of branching, and are composed of various monosaccharides (heteropolysaccharides) and functional groups, such as -OH, -OCH_3_, and -COCH_3_ [6]. Additionally, the solubility of hemicellulose varies significantly. Natural hemicellulose, due to the presence of one or two free hydroxyl groups, typically exhibits hydrophilicity, while synthetic hemicellulose is usually hydrophobic [7]. These characteristics limit its industrial applications.

These shortcomings can be overcome through modification methods, such as etherification, esterification, and cross-linking of hydroxyl groups [8], or by physically altering the microstructure within the hemicellulose system [9]. The main and side chains of hemicellulose contain many free hydroxyl groups, which are ideal sites for chemical modification. Through modification treatments, the physicochemical properties of hemicellulose (such as molecular weight, crystallinity, solubility, surface tension, bioactivity, thermal stability, and rheological properties) can be altered [5]. Therefore, the modification of hemicellulose from grain and oil by-products provides the necessary conditions to fully develop its potential economic value and offers broad application prospects as a new degradable and environmentally friendly material.

Previous reviews on the high-value utilization of GOBPs have mainly focused on the development of functional foods [10,11,12,13,14], bioactive compounds [3,15,16,17,18], and environmental sustainability [3,19,20,21,22]. Lignocellulosic materials include grasses, hardwoods, softwoods, straw, bamboo, bagasse, coconut shells, wood chips, and sawdust [23]. The hemicellulose yields from these sources are as follows: grasses 20–35%, hardwoods 25–35%, softwoods 18–35%, straw 23–36%, bamboo 24–34%, coconut shells 24–28%, and wood chip and sawdust 15–35% (Table 1). 

Hemicelluloses from different sources have different chemical structures and modification effects [29]. Review studies on hemicellulose modification usually focus on hemicellulose from all lignocellulosic materials [5,8,30], with few reviews focusing solely on hemicellulose from GOBPs. This review specifically focuses on hemicellulose sourced from GOBPs, aiming to summarize the modification methods of hemicellulose from GOBPs. Then, it introduces the applications of modified hemicellulose from GOBPs. Finally, the review summarizes the limitations of hemicellulose modification from GOBPs. We hope this review can provide suggestions for high-value applications of GOBPs.

## 2. Modification of GOBPs

### 2.1. Chemical Modification of GOBP

Chemical modification can improve the physicochemical properties of GOBP hemicellulose, including molecular weight, crystallinity, solubility, surface tension, bioactivity, thermal stability, and rheological properties. The purity of hemicellulose has a significant impact on the effectiveness of chemical modification. Low-purity hemicellulose contains impurities, such as other polysaccharides, proteins, or lignin, which compete with or hinder the contact of the modifying agents with hemicellulose molecules. High-purity hemicellulose provides more reactive sites, such as free hydroxyl groups, allowing for a more complete chemical modification [31,32]. Common chemical modification methods include etherification, esterification, and copolymerization [33,34]. Evaluating the effects of modification typically involves assessing physicochemical properties (such as solubility, thermal stability, and mechanical properties) and bioactivity (such as biocompatibility, functionality, antioxidant, and antibacterial properties). These properties largely depend on the degree of substitution (DS) [35].

#### 2.1.1. Etherification

Etherification reactions can significantly increase the molecular weight of hemicellulose, enhance its mechanical strength, and improve the water solubility and thermal stability of GOBP hemicellulose, thereby expanding its application in fields such as food, medicine, and water treatment. By introducing different etherifying agents (such as halides, epoxides, and olefinic monomers), various types of etherification reactions can be carried out, including carboxymethylation, methylation, benzylation, and quaternization (Figure 2) [5]. 

The carboxymethylation reaction is based on the Williamson ether synthesis, where the alcohol groups of 1° and/or 2° polysaccharides can be carboxymethylated (Figure 2A) [36]. Hemicellulose typically undergoes carboxymethylation in a mixture of high-concentration alkaline solution and organic solvents [37]. The DS of the final product is significantly influenced by temperature, reactant ratio, and reaction time, with the optimal DS value obtained only under specific conditions. Peng, Ren, Zhong, Cao, and Sun [32] achieved the highest DS (1.02) under the reaction conditions of a NaOH/chloroacetic acid/wheat straw hemicellulose molar ratio of 1:4:1, microwave heating at 65 °C, and a reaction time of 20 min. Changing the reaction conditions from this baseline resulted in a significant decrease in DS, resulting in more entanglements of carboxymethyl hemicellulose in the solution. Similar results were obtained by other researchers. De Mattos et al. [38] obtained the maximum DS (1.24) for corn fiber xylan under the conditions of a chloroacetic acid and 25% NaOH ratio of 1:4:4.1. The reaction medium is also a critical factor during hemicellulose carboxymethylation. These are usually organic solvents that can disperse and stabilize the reaction system, allowing the alkali and sodium chloroacetate to come into more uniform contact with the hemicellulose [37,39]. Velkova et al. [40] pointed out that the hydrophilicity change in carboxymethylated oat xylan is mainly related to the amount of carboxyl groups introduced, the increase in surface roughness, changes in surface free energy, and changes in molecular weight. By introducing approximately 2 mmol/g of carboxyl groups into the molecular structure of oat xylan, the surface hydrophilicity of the material can be increased by 40%. Spasojevic et al. [41] conducted carboxymethylation treatments on corncob xylan, significantly improving the solubility of xylan. The concentration of ionizable groups in carboxymethyl xylan increased from 1.5 mmol/g to 5.4 mmol/g, which can form stable hydrogels. The solubility was further improved, and it exhibited good enzyme activity retention and operational stability. This makes it highly promising for applications in drug delivery, tissue engineering, and biosensors. The efficiency of the carboxymethylation reaction and the mechanical strength of the modified materials are influenced by the properties of hemicellulose. Zhao et al. [42] explored the carboxymethylation efficiency of corn cob hemicellulose and its impact on enhancing paper strength using different extraction methods. The study found that hemicellulose extracted with 75% ethanol was more readily carboxymethylated due to the higher proportion of branched chains, making the carboxymethylation reaction easier. Hemicellulose extracted with 30% ethanol, having relatively high molecular weight and moderate branching degree, contributed to stronger interactions between pulp fibers, thus showing the best performance in enhancing paper mechanical properties. 

Hemicellulose methylation involves introducing methyl groups onto the hydroxyl groups of hemicellulose, affecting its solubility, swellability, and biodegradability (Figure 2B). Fang et al. [43] prepared and characterized methylated hemicellulose from wheat straw using methyl iodide, sodium hydride, and DMSO as the reaction medium under alkaline conditions, achieving a DS of 1.7 and a reaction yield of 85.1%. The study faced limitations, such as a DS lower than the theoretical maximum, possibly due to degradation of some hemicelluloses during the reaction and incomplete swelling in DMSO. Possible improvements include using microwaves and ultrasound to facilitate methylation [44] and using glycerol to enhance polysaccharide solubility in DMSO [45]. Yang et al. [46] focused on the methylation of soy hull hemicellulose and its impact on the structure and emulsifying abilities, finding that increasing the degree of methylation (DM) from 41.48% to 58.30% enhanced SHP’s surface hydrophobicity and emulsion stability, highlighting its potential in industrial applications, especially in food science.

Benzylation is an important reaction method for enhancing the water resistance of biopolymers. Benzylated polysaccharide materials generally exhibit better plasticity and stronger mechanical properties [34]. Hemicellulose benzylation is a typical Williamson reaction, where the hydrogen atoms in the hydroxyl groups of the hemicellulose chain are replaced by benzyl groups from a benzylating agent, such as benzyl chloride (Figure 2C) [47]. Junli, Xinwen, Linxin, Feng, and Runcang [47] prepared novel hydrophobic hemicellulose with DS from 0.09 to 0.35 by benzylating wheat straw hemicellulose using benzyl chloride in an ethanol/water system with sodium hydroxide as a catalyst. The benzylation reaction results were influenced by the ratio of ethanol to water, the molar ratio of reactants, temperature, and reaction time. In the study by Kaur and Kaur [48], by adjusting the temperature, time, and reagent concentrations, the degree of benzylation of rice straw xylan could be increased while avoiding side reactions, resulting in improved thermal stability, antioxidant properties, and hydrophobicity of the hemicellulose.

Quaternized hemicellulose, characterized by its cationic properties, enhances water solubility and surface activity, finding broad applications in water treatment, drug release, and food industries; trimethylamine chloride is the most commonly used quaternizing agent (Figure 2D) [49]. By increasing the positive charge density of molecules, the quaternization process improves the interactions with negatively charged substances [50]. The introduction of cationic groups also imparts certain antibacterial properties to the material, making quaternized hemicellulose widely used in the food and biomedical fields [51]. For example, Fröhlich et al. [52] found that modified corn cob xylan, especially quaternized xylan, showed significantly improved antibacterial and antioxidant activities due to the introduction of quaternary ammonium groups with strong antioxidant activity. 

#### 2.1.2. Esterification

The esterification modification of hemicellulose involves the reaction of anhydrides, carboxylic acids, or acyl halides with the hydroxyl groups on the main chain of hemicellulose under certain conditions to form ester bonds [34], and esterification of hemicellulose can also inhibit the formation of strong hydrogen-bonded networks, thereby improving the flexibility of hemicellulose-based materials [30,53]. Acetylation is the most common method of esterification, but other methods, such as oleoylation, have also been applied in the esterification of hemicellulose from GOBPs (Figure 3).

In the acetylation reaction, the hydroxyl groups in hemicellulose are converted into acetyl groups through esterification, which can alter the spatial arrangement of polysaccharide chains, exposing hydroxyl groups and changing the solubility and hydrophobicity of hemicellulose [54]. Acetylation reactions of hemicellulose can be categorized into solvent methods and alkaline methods, depending on the reagents used. The solvent method requires dissolving hemicellulose in an organic solvent (such as formamide, DMAC/LiCl, DMSO, and pyridine), stirring, and heating. Then, the main acetylating reagent, such as acetic anhydride, is mixed with a catalyst and added to the reaction system [55]. Egüés et al. [56] used a pyridine/acetic anhydride system to acetylate purified corn cob xylan, achieving a DS of about 1.9, which significantly improved the water resistance and mechanical properties of the material. The properties of the acetylated products are also related to the proportion of arabinose side chains in the hemicellulose. Hydroxyl groups usually come from the side chains; a lower proportion of side chains can inhibit the acetylation of hemicellulose, increasing the hydrophobicity of the product [57]. Yilmaz-Turan et al. [58] found that the sugar content and molecular weight of wheat bran hemicellulose decreased after acetylation in a DMSO/*N*-methylimidazole (NMI) system, possibly due to the breakage of polysaccharide chains. The properties of the acetylated products are related to the purity of the hemicellulose; higher purity hemicellulose tends to achieve a higher DS, which correlates with better thermal and mechanical properties. The catalyst is an important factor in the solvent method of hemicellulose acetylation. Fang et al. [59] acetylated wheat straw hemicellulose using 4-dimethylaminopyridine as a catalyst, achieving DS values between 0.59 and 1.25. Under optimal conditions (85 °C, 60 h), approximately 75% of the free hydroxyl groups in hemicellulose were acetylated. Sun, Min, and Sun [31] used pyridine as a catalyst to esterify wheat straw hemicelluloses. The optimal DS was 1.67. Although pyridine is the most commonly used catalyst, it is toxic [60]. Ionic liquids (ILs) with iodine are an alternative. In the acetylation of wheat straw hemicellulose, Ren et al. [61] used 15% iodine in the ionic liquid 1-butyl-3-methylimidazolium chloride ([Cmim]Cl), and about 83% of the hydroxyl groups in wheat straw hemicellulose were acetylated. Stepan et al. [62] achieved complete acetylation of rye arabinoxylan in the novel protic ionic liquid 1,5-diazabicyclo[4.3.0]non-5-enium acetate ([DBNH][OAc]), with 1,5-diazabicyclo[4.3.0]non-5-ene (DBN) as the catalyst; the obtained DS was about 2. However, ionic liquids are expensive [63]. *N*-Bromosuccinimide (NBS) is an inexpensive and efficient catalyst for the acetylation of hemicellulose. Xu et al. [64] used NBS as a catalyst to acetylate wheat straw hemicellulose under microwave conditions, achieving a DS of 1.34. The alkaline acetylation method involves first dissolving the original polysaccharide in water, adjusting the pH of the solution to 8.0–9.0 by adding NaOH, and then gradually adding the acetylating reagent, such as acetic anhydride, while stirring. For example, Sun et al. [65] studied the esterification reaction of wheat straw hemicelluloses with succinic anhydride (SA) in an alkaline aqueous system. The best results were obtained under the conditions of pH 8.5–9.0, temperature 25–28 °C, reaction time 1–2 h, and a molar ratio of succinic anhydride to hemicelluloses of 1:1. The DS ranged from 0.017 to 0.21. Thermal gravimetric analysis (TGA) and differential scanning calorimetry (DSC) results showed that the thermal stability of the esterified polymer increased with chemical modification. The DS in aqueous acetylation reactions is usually lower. However, aqueous acetylation offers advantages, such as reduced health hazards, lower environmental pollution, and lower costs [55].

The characteristic of hemicellulose oleoylation is the replacement of hydroxyl groups with hydrophobic oleoyl groups (Figure 3). Sun et al. [66] conducted the oleoylation of wheat straw hemicellulose in a homogeneous *N*,*N*-dimethylformamide/lithium chloride (DMF/LiCl) system at 75 °C for 35 min, achieving 90% oleoylation of the hydroxyl groups in wheat straw hemicellulose. This homogeneous system not only improved the reaction efficiency but also ensured uniformity and high quality of the product. The oleoylated hemicellulose exhibited better hydrophobicity and thermal stability.

Lauroylation modification involves introducing lauroyl groups (C_12_H_23_CO-) into polymer molecules, typically achieved by reacting with lauroyl chloride (C_12_H_23_COCl). Ren et al. [67] lauroylated wheat straw hemicellulose in a DMF/LiCl system using 4-dimethylaminopyridine (DMAP) as a catalyst, obtaining a product with a DS of 1.63 at 78 °C within 5 min. Similarly, Peng et al. [68] achieved a DS of 1.54 at 40 °C within 35 min in the same system. The study found that the thermal stability of lauroylated hemicellulose increased with the degree of substitution, and conducting the reaction at lower temperatures (40 °C) reduced product degradation.

In other types of esterification reactions, Li [69], for example, used four kinds of hydroxycinnamic acids (FA, p-CA, SA, CA) to modify arabinoxylan (CAX) extracted from corn bran through esterification reactions. The study found that CA-CAX-0.32, which contains more phenolic hydroxyl groups and hydrogen ions, exhibits antioxidant properties and can inhibit oxidative damage to cells caused by free radicals.

#### 2.1.3. Copolymerization

Copolymerization modification of hemicellulose is a simple, flexible, and efficient way to significantly improve its physicochemical properties, resulting in derivatives with excellent performance. This modification can be tailored to hemicellulose’s characteristics, adjusting the polymerization method, monomers, graft uniformity, and density, and even adding different functional groups based on application needs [70]. Under the action of initiators, the free hydroxyl groups on the hemicellulose molecular chain can undergo graft polymerization with monomers. The grafted monomers can include vinyl chloride, acrylonitrile, acrylamide, and others [8]. Hemicellulose graft copolymers combine the inherent excellent properties of hemicellulose with the new characteristics imparted by the synthetic polymer branches [71]. Fanta et al. [72] grafted acrylonitrile and methyl methacrylate onto low-lignin wheat straw hemicellulose, using ceric ammonium nitrate as the initiator. The study achieved grafting of acrylonitrile and methyl methacrylate onto hemicellulose with a maximum grafting efficiency of 37%, producing materials with improved water absorption and thickening properties. In addition to ceric ammonium nitrate, other oxidizing agents can also serve as initiators for graft polymerization reactions. Soliman [73] used potassium permanganate to initiate the copolymerization of corn cob hemicellulose with acrylonitrile. The grafting efficiency of the hemicellulose graft copolymer was primarily dependent on the monomer concentration, initiator amount, reaction time, and reaction temperature. In addition to these factors, the physicochemical properties of hemicellulose also affect copolymerization. Peroval et al. [74] grafted stearyl acrylate and stearyl methacrylate onto corn bran arabinoxylan using oxygen plasma pretreatment and electron beam irradiation grafting techniques. The oxygen plasma pretreatment made the material surface more reactive, facilitating subsequent grafting. The electron beam irradiation resulted in smaller particles and a uniform distribution, which helped form a denser film structure, significantly enhancing the film’s surface hydrophobicity and water vapor barrier properties. Littunen et al. [75] pointed out that the high molar mass and high solubility of three types of hemicellulose (oat, bleached oat, wheat) favor the copolymerization reaction, while the high lignin content may hinder the copolymerization process, making it difficult to form dense hydrophobic films. Littunen et al. [76] used two α-arabinofuranosidases, SthAbf62A from Streptomyces thermoviolaceus and AXHd3 from Bifidobacterium sp., to remove α-arabinofuranose (Araf) and obtain high molecular weight and high degree of polymerization of xylan (Figure 4). The enzymatic debranching treatment could increase the grafting yield, and the debranched xylan formed a denser and less hydrated adsorption layer on the cellulose surface. Svärd et al. [77] extracted high-molecular-weight xylan from rapeseed straw using 1 M NaOH at 80 °C and 110 °C. The xylan was esterified with glycidyl methacrylate (GMA) to introduce vinyl groups, followed by radical polymerization to graft octadecyl acrylate (ODA) onto the xylan. The modified xylan was mixed with polycaprolactone (PCL), enhancing the material’s rigidity while maintaining good ductility. The Young’s modulus increased from 300 MPa for pure PCL to 440 MPa for PCL with 20% modified xylan, while the elongation at break remained at over 500%. The thermal stability was significantly improved, with modified xylan starting to degrade at 300 °C compared to 200 °C for unmodified xylan. Through chemical modification and composite material technology, grain and oil by-products can be effectively utilized to develop commercially valuable biodegradable materials, reducing the dependence on petroleum-based plastics and providing significant environmental and economic benefits. 

Media have a significant impact on the polymerization reaction and product performance. Ünlü et al. [78] grafted polyacrylonitrile (PAN) onto xylan extracted from corn cobs to prepare xylan-polyacrylonitrile graft copolymers (X-PAN) using ceric ammonium nitrate (CAN) as the initiator. They compared two different media: nitric acid (HNO_3_) and perchloric acid (HClO_4_). The results showed that in the nitric acid medium, the optimal conditions were 50 °C, 1 h, CAN concentration of 21.7 mmol/L, nitric acid concentration of 0.1 mol/L, xylan concentration of 0.2% (*w*/*v*), and acrylonitrile concentration of 0.49 mol/L, achieving a conversion rate of 96% and a molecular weight of 249 kDa. In the perchloric acid medium, the optimal conditions were 27 °C, 1 h, CAN concentration of 5.4 mmol/L, perchloric acid concentration of 0.2 mol/L, xylan concentration of 0.5% (*w*/*v*), and acrylonitrile concentration of 0.82 mol/L, achieving a conversion rate of 83% and a molecular weight of 271 kDa. The two products exhibited different FTIR characteristics and thermal behaviors. Li and Zhou [79] successfully prepared hemicellulose-g-polyacrylamide copolymers through a radical copolymerization reaction in a dilute alkaline water medium, significantly improving the solubility and thermal stability of corn cob hemicellulose. Compared to organic solvents, aqueous media are more environmentally friendly.

### 2.2. Physical Modifications

Physical modification can cut the main chain or side chains of xylan, maintaining the basic structure of polysaccharides to alter the microcomposition and structure within the hemicellulose system, causing only some conformational changes and improving its functionality and processing performance in various applications [51]. The most commonly used methods include heat treatment, ultrasonic treatment, and mechanical treatment [80,81].

#### 2.2.1. Extrusion

Extrusion processing technology is typically a combination of thermal and mechanical treatments to break and degrade xylan molecular chains under high shear force and high temperature, leading to changes in the physicochemical properties and molecular redistribution of hemicellulose [82]. Fang et al. [83] pointed out that during the extrusion process, wheat xylan with low branched structures exhibits high viscosity characteristics in flow behavior, making it suitable for industries such as food, pharmaceuticals, and cosmetics. The ratio of arabinose to xylose is negatively correlated with the extrusion speed. The weight distribution of arabinoxylan in rice bran can be optimized to obtain more low-molecular-weight arabinoxylan with immunomodulatory activity, thereby enhancing its application value in food and health products [82]. After extrusion, the water solubility and antioxidant activity of xylan are enhanced [84,85]. However, the properties of hemicellulose after extrusion varied from different processing conditions and raw material characteristics. Demuth et al. [86] found that both extrusion and grinding treatments resulted in decreased antioxidant activity, with the extrusion-treated samples showing the lowest antioxidant activity, since high temperature and high shear force during extrusion led to the degradation or oxidation of phenolic compounds.

#### 2.2.2. Ultrasonic

Ultrasonic modification can break certain chemical bonds in xylan macromolecules, as hemicellulose’s main chain, reducing the molecular weight and enhancing the water solubility and antioxidant activity of hemicellulose [51]. Li et al. [87] studied the modification effects of ultrasonic treatment, which increased the release of phenolic compounds on wheat bran arabinoxylan, thereby enhancing the radical scavenging and reducing abilities of arabinoxylan. The reduction in molecular weight affected the oxidative cross-linking capacity of arabinoxylan, with higher molecular weight arabinoxylan more likely to form highly elastic and viscous gel structures.

#### 2.2.3. Physicochemical Modifications and Enzymatic Hydrolysis

Physical modification is often combined with chemical treatments. For example, in the study of Vandenbossche et al. [88], sweet corn cobs, oil palm empty fruit bunches, and barley straw were used as raw materials; alkaline pretreatment was carried out in a twin-screw extruder, followed by neutralization with phosphoric acid. Enzymes were then added during the extrusion process to initiate the saccharification of hemicellulose and cellulose, providing efficient raw materials for biofuel production. Significant differences were found in the behavior of the three biomasses during pretreatment and enzymatic hydrolysis due to their different chemical compositions and physical structures. The enzymatic hydrolysis efficiency of sweet corn cobs was the highest (26%), while that of oil palm empty fruit bunches was the lowest (8%). Chen et al. [89] separated the lignocellulose components from corn husk using ultrasonic-assisted mild alkaline treatment and deep eutectic solvent (DES) treatment (Figure 5). The extracted hemicellulose included water-soluble macromolecular hemicellulose with a molecular weight of 25.21 kDa and water-insoluble hemicellulose that formed a stable nanoparticle dispersion after mechanical homogenization. The separated nanocellulose and nanofibrillated cellulose were synergistically applied for the first time to modify marine bio-based sodium alginate films. The study found that ultrasonic-assisted mild alkaline treatment effectively inhibited the dissolution of lignin. The composite films exhibited higher tensile strength and elongation at break, reaching up to 98 MPa and 4.5%, which were 2.2 times and 1.7 times higher than those of pure sodium alginate films, respectively. The gas barrier properties of the composite films were also significantly enhanced, with water vapor and oxygen permeability reduced by approximately 46% and 83%, respectively. Nanocellulose and nanofibrillated cellulose filled the voids in the alginate films and interacted with sodium alginate through hydrogen bonds, forming a dense network structure that reduced free volume and hindered gas diffusion through the films. 

## 3. Applications of GOBP Hemicellulose

### 3.1. GOBP Hemicellulose in Film Production

In recent years, significant progress has been made in developing high-performance hemicellulose-based films. Hemicelluloses, as natural biopolymers, have garnered considerable attention in the domain of environmental materials owing to their exemplary biodegradability [90], which is capable of rapid decomposition under natural conditions, significantly mitigating the environmental impacts when juxtaposed with traditional plastics [91]. However, due to the structural properties of hemicellulose, there are challenges in its compatibility with traditional plastics, as well as in the thermal stability, mechanical properties, and water vapor permeability of hemicellulose materials [81]. Xylan, with a more complete structure, has a higher elongation at break. Svärd et al. [92] used optimized hydrothermal alkaline extraction conditions to extract high-purity hemicellulose, primarily consisting of galactoglucomannan and xylan, from rapeseed straw. The film achieved an elongation at break exceeding 60%. Enhancing the performance of films can be effectively achieved through the addition of plasticizers. Azeredo et al. [93] used citric acid (CA) as a plasticizer to promote ester bond formation between CA and wheat straw hemicelluloses; this process resulted in improved water resistance and water vapor barrier properties of the films. Kocabaş et al. [94] used a combination of acid hydrolysis and mechanical defibrillation techniques with sodium alginate and added CA as a plasticizer. The extracted hemicellulose and cellulose from barley bran could form bio-composite films with excellent mechanical properties and gas barrier performance. The study results indicated that the addition of CA not only acted as a plasticizer but also improved the flexibility and processability of the material by forming ester bonds with the hemicellulose chains. Kapil et al. [95] chemically modified rice straw xylan into acetylated xylan (AX) and carboxymethylated xylan (CMX) and created bio-composite films with polyvinyl alcohol (PVA) and CA. The introduction of AX and CMX enhanced the mechanical strength of the bio-composite films. In CMX bio-composite films, the unreacted CA contained undissociated carboxylate ions (COO-) that could kill bacteria by disrupting their cell membranes. However, high concentrations of AX and CMX could lead to uneven intermolecular cross-linking, reducing the elongation at break of the films. The acetylation process reduced the number of free hydroxyl groups, thereby decreasing the antioxidant activity [95]. This suggests the need to prepare different modified hemicellulose-based materials according to specific requirements.

Hemicellulose and cellulose nanofiller composites have gained significant attention. Pereira et al. [96] innovatively modified wheat straw hemicellulose films by incorporating varying concentrations of cellulose nanocrystals (0–8 wt%) and citric acid (0–30 wt%). They found that the optimal addition of 5.9 wt% CNC and 30 wt% citric acid significantly enhanced the film’s physical properties; the enhancement included improvements in tensile strength, water vapor barrier properties, and water resistance. The research of Xu et al. [97] utilized Artemisia selengensis straw to extract hemicellulose and cellulose nanocrystals, creating enhanced composite films with polyvinyl alcohol, resulting in a substantial increase in tensile strength by 80.1% to 36.21 MPa with a 9% inclusion of cellulose nanocrystals; additionally, the water vapor transmission rate of these films decreased by 15.45% when enhanced with 12% cellulose nanocrystals. These advancements suggest the potential use of these films in applications like biomedicine packaging materials and humidity sensors. Hemicellulose nanocrystals and nanocellulose filled the pores of the films, forming a dense structure that significantly enhanced the mechanical properties, thermal stability, and gas barrier performance of the composite films. The presence of hemicellulose nanocrystals can somewhat slow down the degradation process of the material; however, due to their inherent biodegradability, the overall material can still naturally degrade in the environment. Rice straw hemicellulose was processed into nanoparticles with an average particle size of 141 nm through acid hydrolysis and intense ultrasonic treatment. The intense ultrasonic treatment caused the breaking of cellulose and hemicellulose molecular chains, forming spherical nanoparticles. These nanoparticles exhibited good dispersibility and stability and were used to enhance starch-based composite films. This significantly improved the mechanical properties, gas barrier performance, and thermal stability of the films while maintaining good biodegradability [98].

### 3.2. GOBP Hemicellulose in Hydrogel Formation

Compared to other lignocellulosic biomasses (cellulose, lignin), hemicelluloses are characterized by their low molecular weight, low degree of polymerization, and high degree of branching [99]. They also possess numerous active hydrophilic functional groups (hydroxyl, carboxyl), making them easily soluble and modifiable, thus serving as excellent natural substrates for developing functional biomass-based hydrogels [100]. However, their low molecular weight and degree of polymerization can impact the strength and usage stability of hemicellulose-based hydrogels [80]. Researchers have conducted in-depth studies to improve the strength and enrich the functionalities, categorizing hemicellulose-based hydrogels into physically and chemically cross-linked types [6].

Physical cross-linking refers to the formation of physical cross-linking points between macromolecular chains under physical conditions, such as heating, high pressure, freezing, irradiation, and ultrasound, through hydrogen bonding, hydrophobic interactions, host–guest interactions, electrostatic effects, polymer chain entanglement, and crystallization [80,101,102]. Meena et al. [103] studied the properties of mixed hydrogels of κ-carrageenan (kC) and oat bran xylan (OSX). The mixed gels, with 50–90% (by weight) OSX, showed significantly reduced segregation in kC gels and a notable increase in swelling capacity. The gels with kC/OSX50-90% had lower gelation and melting temperatures. The addition of OSX formed a more stable interpenetrating network structure, enhancing the stability of the hydrogels. Higher solution concentrations increase the physical cross-linking points for denser and more stable gel structures [102]. Talantikite et al. [104] mixed cellulose nanocrystals (CNC) with arabinoxylan (AX) to synthesize fully bio-based hydrogels with tunable mechanical properties (Figure 6). The amount of AX adsorbed on the surface of CNCs was positively correlated with the AX concentration. 

Chemical cross-linking typically involves the initiation of hydroxyl groups in hemicellulose to produce oxy radicals, which undergo free radical graft copolymerization with polymer molecular chains, ultimately forming a three-dimensional network structure under heat, light, radiation, or cross-linking agents [5]. Graft copolymerization enhances the molecular weight and thermal stability of hemicellulose, introducing new functional groups to improve the strength and expand the functionalities [5]. As illustrated in Figure 7, Sun et al. [105] introduced acrylic acid (AAc) into the hemicellulose structure through a radical copolymerization reaction, successfully preparing pH-sensitive hydrogels based on wheat straw hemicellulose.

The acrylic acid chains formed cross-links with the hemicellulose backbone. These hydrogels exhibited significant swelling differences under various pH conditions. In acidic environments, most carboxyl groups remained non-ionized, resulting in low charge density, with water diffusion primarily controlled by the concentration gradient. In neutral and alkaline conditions, the ionization of carboxyl groups generated strong electrostatic repulsion, causing the hydrogel to relax and absorb more water, thus exhibiting a higher swelling ratio. The swelling ratio was positively correlated with the drug release rate, indicating the hydrogel’s potential in drug delivery applications. The incorporation of Fe_3_O_4_ nanoparticles imparted magnetic properties to the wheat straw hemicellulose hydrogel. The porous structure and uniformly distributed Fe_3_O_4_ nanoparticles enhanced the adsorption capacity and rate. The magnetic properties of the hydrogel enable it to be easily separated from the solution under the influence of a magnetic field. The prepared hydrogel exhibited significant swelling behavior in aqueous solutions and displayed superparamagnetic characteristics. The saturation magnetization was 4.21 emu/g. At pH 8, the hydrogel achieved an adsorption capacity of 65.28 mg/g for methylene blue, with a removal rate of 95.58%. However, at higher pH values, Na+ ions in the solution occupy the adsorption sites, thereby reducing the adsorption efficiency [106]. Wang et al. [107] studied the molecular characteristics and gelation ability of functional hydrogels formed by oxidative cross-linking of arabinoxylan extracted from three types of wheat bran. The study found that high molecular weight and high ferulic acid content contribute to the formation of hydrogels with good elasticity, and high concentrations of arabinoxylan can form stable gel networks more quickly. Li et al. [108] used carboxymethylated arabinoxylan (CMAX) from wheat bran, which was cross-linked with Fe3+ ions through the carboxyl groups (COO−) of CMAX using a reverse emulsion polymerization method to form microgels. The resulting CMAX microgels were regular spherical particles with sizes ranging from 30 to 80 μm. These microgels exhibited high stability under acidic conditions (pH 2.2) and significant swelling behavior under neutral conditions (pH 7.0). This pH sensitivity allows CMAX microgels to exhibit excellent controlled release characteristics in simulated gastrointestinal environments, making them suitable as delivery carriers. Ref. [109] synthesized chemically cross-linked composite hydrogels based on acetylated corn cob xylan and silanized oxidized graphene. The acetylation and silanization modifications of graphene oxide (GO) and xylan introduced hydroxyl and carboxyl functional groups into the hydrogels, aiding in the adsorption of metal ions.

Enzymatic reactions are a popular method for preparing hydrogels, as shown in Figure 8. During the cross-linking process of xylan (especially arabinoxylan), laccase oxidizes ferulic acid (FA) residues to generate phenoxy radicals. Through coupling reactions, di-ferulic acid (di-FA) and tri-ferulic acid (tri-FA) are formed, resulting in the formation of numerous covalent bonds between the molecular chains of arabinoxylan, particularly phenol–phenol bonds (such as 5-5′, 8-5′, 8-O-4′ di-ferulic acid structures). This initiates the polymerization reaction, forming a cross-linked network [51].

Chimphango et al. [110] prepared oat bran xylan hydrogels through selective enzymatic hydrolysis. Recombinant *α*-l-arabinofuranosidase (AbfB) was used to selectively remove arabinose side chains from oat bran xylan. The removal of arabinose reduced steric hindrance between molecules, promoting the aggregation of xylan molecules and hydrogel formation. Zhang et al. [111] extracted arabinoxylan (CAX) from corn bran using an alkaline solution and cross-linked it through laccase treatment. At low pH, the carboxyl groups on the arabinoxylan chains were protonated, promoting hydrogen bond formation and reducing repulsion between polymer chains, thus facilitating gel formation. Higher ferulic acid content resulted in a higher degree of cross-linking, increasing the strength of the formed SCCAX gels. The authors also noted that the gel formation mainly relied on reversible non-covalent interactions. The increased carboxyl and ferulic acid groups on the arabinoxylan chains after cross-linking added negative charges, contributing to the gel’s formation and stability. SCCAX gels have broad applications in food and drug delivery systems, particularly in low-sugar, low-pH gels or applications requiring gel formation in the stomach to delay gastric emptying and increase satiety. Spasojevic, Prokopijevic, Prodanovic, Zelenovic, Polovic, Radotic, and Prodanovic [41] prepared tyrosine carboxymethyl corn cob xylan (Tyr-CMX) hydrogels through cross-linking reactions with horseradish peroxidase (HRP) and hydrogen peroxide. Starch was immobilized in Tyr-CMX hydrogel microbeads via emulsion polymerization. The microbeads formed by emulsion polymerization had smaller diameters, reducing diffusion limitations and improving enzyme activity retention. The introduction of high-functional groups and smaller microbead diameters reduced enzyme leakage, enhancing operational stability and reusability of the enzyme. Li et al. [112] found that subcritical water extraction effectively extracted high molecular weight and high ferulic acid content AX, retaining more ferulic acid esters, which could significantly enhance its oxidative cross-linking ability. They then used laccase to cross-link corn bran arabinoxylan (AX) to form gels. Chen [29] also used a subcritical water extraction and laccase-induced oxidative cross-linking method to prepare hydrogels from arabinoxylan of corn and rye bran. The study pointed out that ferulic acid content and molecular weight are the key factors determining the cross-linking performance of arabinoxylan. Munk et al. [113] treated glucuronoxylan-rich arabinoxylan extracted from corn bran with laccase from Myceliophthora thermophila. The laccase catalyzed the oxidative cross-linking of ferulic acid groups to form covalent di-ferulic acid cross-links, producing strong hydrogels. Li [69] used laccase (derived from Trichoderma and oyster mushrooms) as a cross-linker to cross-link hydroxycinnamic-acid-esterified corn bran arabinoxylan at 25 °C. The author noted that methoxy and hydroxyl groups provide a higher cross-linking density, and the hydrogels exhibited good mechanical properties and biocompatibility, with broad application prospects in antioxidant skincare products and functional foods. Yilmaz-Turan et al. [114] extracted arabinoxylan (FAX) from wheat bran and formed functional hydrogels through laccase oxidative cross-linking. Laccase induced the conversion of ferulic acid (FA) units into various dimer forms, increasing the crystallinity and molecular weight of FAX and allowing FAX chains to pack more tightly. The hydrogels could be reused by freeze-drying and re-suspension. Freeze-drying converted the water molecules in the hydrogels directly into gas through sublimation, avoiding the presence of liquid water, forming a porous dry structure that facilitated rapid water absorption and restored the swelling properties of the hydrogels. Re-suspension in specific pH buffers strengthened the chemical bonds and physical interactions within the hydrogels, restoring the original structure of the hydrogels.

The development trend of hemicellulose-based hydrogels focuses on enhancing mechanical properties, incorporating multifunctionality, and expanding potential applications. The innovations primarily involve the combination of hemicelluloses with various other components to improve strength, self-healing capabilities, and functionality [115]. The methods have evolved to include physical cross-linking, chemical modifications, and the utilization of nanoparticles, indicating a shift toward creating materials with specific attributes, such as conductivity, biodegradability, and suitability for biomedical and environmental applications [101]. This progression signifies a growing interest in sustainable and versatile materials that can be customized according to diverse application needs.

### 3.3. GOBP Hemicellulose in Three-Dimensional (3D) Printing

Currently, there is great interest in exploring natural and renewable polymers for manufacturing various 3D printed products [116]. Hemicellulose, with its environmentally friendly, biodegradable, renewable, and biocompatible properties, has shown significant potential in 3D printing applied in various fields, such as microelectronics, photovoltaics, energy, tissue engineering, and food manufacturing [117]. Bahcegul et al. [118] developed a 3D printing technology based on corn cob hemicellulose. The hemicellulose was prepared using a 10% KOH solution for extraction, followed by acetic acid and ethanol precipitation. The study found that the viscosity of the slurry was highly sensitive to changes in water content and temperature. Adding NaOH reduced the viscosity of the slurry, making the printing process smoother and more continuous. The optimal printing conditions were found to be 65% water content, 80 °C printing temperature, and an extrusion multiplier of 1.5. Although the mechanical properties of the 3D printed materials were not as good as those produced by traditional methods, the ease of processing and cost advantages made this approach competitive for specific applications. Bahcegul et al. [119] developed a more environmentally friendly method, using a 5% KOH solution to extract hemicellulose from corn cobs and forming a gel by evaporating the water, without the need for further purification. The resulting product exhibited similar mechanical strength to the product from the previous study [118].

Although 3D printing technology provides a new avenue for the processing of hemicellulose and its derivatives, research in this aspect remains incomplete. Firstly, using hemicellulose and its derivatives directly as raw materials for 3D printing biomaterials is extremely challenging. This is due to the less than ideal mechanical properties, stability, and resolution of 3D printed products made from hemicellulose materials [120]. Therefore, it is necessary to adjust the composition of composite materials and adapt different 3D printing techniques. Additionally, the cost effectiveness and environmental impact are also important considerations. Traditional solvents used in preparing 3D printing materials can be harmful or expensive; therefore, there is a need to research environmentally friendly and cost-effective solvents.

### 3.4. GOBP Hemicellulose in Adsorbent Production

With industrial development, the threat of harmful pollutants is increasingly serious, and effectively removing these toxic chemicals has become a global challenge. Among various treatment methods, adsorption is gaining more attention due to its ability to target a wide range of pollutants and relatively low cost. Hemicellulose-based materials, with their unique physicochemical properties, can be used to prepare hydrogels and activated carbon materials. Hydrogels are highly effective in adsorbing heavy metal ions, organic dyes, water and salt solutions, while active carbons are particularly suitable for adsorbing gaseous substances. Mohammadabadi and Javanbakht [121] prepared a hydrogel by mixing barley straw hemicellulose with sodium alginate for the removal of lead ions from aqueous solutions. Under the conditions of pH 5, an adsorbent dosage of 0.4 g/L, an initial lead ion concentration of 210 mg/L, and a temperature of 55 °C, the maximum adsorption capacity reached 277.78 mg/g, with a removal efficiency exceeding 99%. The strong interactions between hemicellulose and calcium alginate formed an adsorbent material with a high specific surface area and porous structure, effectively increasing the adsorption sites for lead ions. Additionally, a regeneration treatment with 0.01 M nitric acid effectively removed lead ions from the adsorbent surface, restoring its activity. After five adsorption–desorption cycles, the adsorbent maintained a high lead ion adsorption capacity. The study of Sun, Xie, Shan, Li, and Sun [109] chemically cross-linked composite hydrogels based on acetylated corn cob xylan and silanized graphene oxide (GO), which were synthesized through radical polymerization. The chemical cross-linking of acetylated xylan and silanized GO formed a stable three-dimensional polymer network, providing numerous adsorption sites. The hydrogel maintained a high Cu^2^⁺ removal rate even after multiple cycles of use, with a desorption rate still reaching 77.3% after six cycles.

Aerogels are also common adsorbent materials. Guan et al. [122] prepared a wheat straw hemicellulose-chitosan aerogel containing Fe_3_O_4_ through oxidation and cross-linking reactions. Fe_3_O_4_ not only imparted magnetic properties to the aerogel but also acted as a cross-linking agent to enhance its structural stability. The magnetic aerogel efficiently adsorbed Congo red dye and could be quickly recovered under an external magnetic field, demonstrating good reusability. Sharma et al. [123] developed a 3D multifunctional fluorescent aerogel (CS@DAHCA) based on rice straw hemicellulose for the ultrasensitive detection and removal of arsenic ions (As(III)) and ciprofloxacin (CPR) from water. By oxidizing with NaIO_4_, the C2 and C3 hydroxyl groups of hemicellulose were converted to aldehyde groups, increasing its reactivity and forming a more stable cross-linked structure. Through Schiff base reactions, DAHC formed a three-dimensional porous network with chitosan, providing numerous active sites. The detection limits for As(III) and CPR were 3.529 pM and 55.2 nM, respectively. This high sensitivity was mainly due to the aerogel’s high surface area and abundant functional groups. CS@DAHCA maintained over 95% adsorption efficiency after 10 cycles, demonstrating excellent regeneration capacity and adsorption performance. Aside from forming hydrogels and aerogels, hemicellulose tends to decompose into H_2_O, CO_2_, and CO during biomass thermal treatment, leading to a structure rich in micropores. Chen et al. [124] developed a mild and straightforward method to decompose lignocellulose from corn straw into cellulose, hemicellulose, and lignin, utilizing these monomers to fabricate three types of porous carbon materials. Through carbonization and chemical activation, the resulting cellulose (PCCC), hemicellulose (PCHC), and lignin (PCLC)-based porous carbons exhibited exceptional surface area and porosity. Notably, the hemicellulose-based porous carbon (PCHC) demonstrated significant adsorption performance, proving the effective strategy of breaking down lignocellulosic biomass into its components for creating enhanced porous carbon materials. This approach not only offers a new pathway for the comprehensive utilization of agricultural waste like corn straw but also contributes fresh insights into the development of materials for water treatment and environmental remediation.

## 4. Limitations of Hemicellulose Modification from GOBPs

Hemicellulose from grain and oil processing by-products is widely used for modification applications due to its availability and cost effectiveness. However, several limitations restrict its broader usage. The main limitations of hemicellulose from GOBPs are outlined below (Table 2).

The complex and heterogeneous composition of grain and oil processing by-products affects the efficiency of the modification process and the quality of the final product [125]. Impurities can lead to suboptimal modification results. Many modification processes require multi-step chemical reactions or complex physical treatments, which are not only time-consuming but also require specific equipment and conditions, increasing the technical difficulty [5]. There is still a lack of industrial-scale, more economical, and sustainable solutions to fully utilize GOBPs [4]. Modified grain and oil by-product materials may perform excellently in some applications but may be restricted in others. Certain modified materials may be unstable under acidic, alkaline, or high-temperature conditions [126]. Although modification can enhance certain properties, the modified materials might exhibit deficiencies in other properties. For instance, acetylation improves meltability but reduces thermal stability [127]. The acetylation process also reduces the number of free hydroxyl groups, thereby decreasing the antioxidant activity of xylan [95]. The modification process typically requires expensive chemical reagents and specialized equipment, leading to high production costs for the modified materials. This can limit their large-scale commercial applications and make it challenging to ensure the economic viability and profitability of hemicellulose modification products from grain and oil processing by-products. Some chemical modification processes can generate harmful by-products or waste, increasing the environmental burden associated with waste treatment. The development of green chemical modification technologies needs further research. Many modification processes require high temperatures, high pressure, and other conditions, leading to significant energy consumption.

## 5. Conclusions

This article summarizes the modification and application of hemicellulose in GOBPs. Due to the increasing scarcity of non-renewable resources, it is crucial to find renewable and environmentally friendly materials. The grain and oil processing industry produces large amounts of by-products, such as bran, straw, and seed hulls, which are low-cost and high-yield, making them ideal raw materials for hemicellulose modification. However, natural hemicellulose has complex structures, many branches, and poor solubility, limiting its industrial applications. Through chemical and physical modifications, the physicochemical properties of hemicellulose can be significantly improved, showing great potential in applications such as film production, hydrogel formation, three-dimensional (3D) printing materials, and adsorbents for environmental remediation. The utilization of hemicellulose in GOBPs faces challenges, such as complex composition, complicated modification processes, high production costs, and significant environmental burdens. To address these issues, optimizing and adopting green modification methods, developing cost-effective industrial technologies, improving material properties, and enhancing interdisciplinary collaboration are essential. These improvements will enable more effective utilization of hemicellulose, achieving high-value applications in environmental materials and biomaterials.

## Figures and Tables

**Figure 1 polymers-16-01750-f001:**
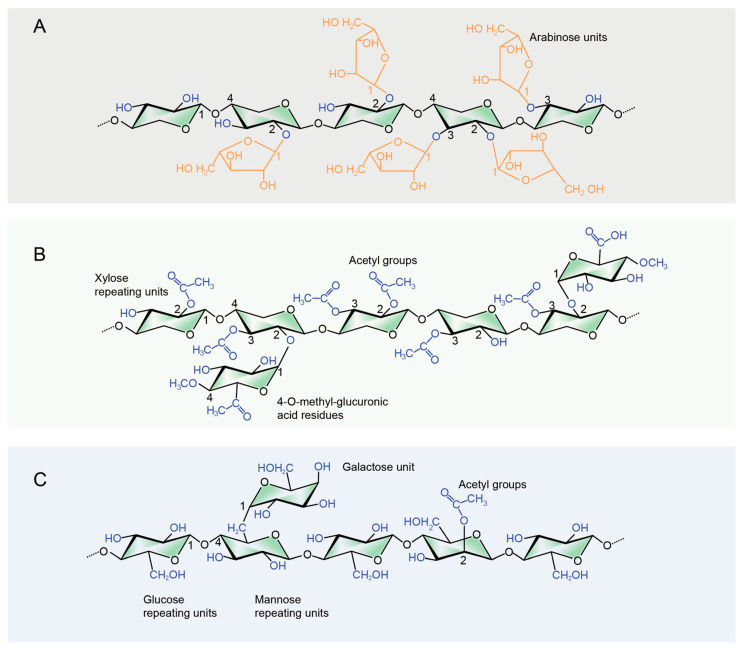
Representative chemical structures of hemicellulose ((**A**): Grasses, (**B**): Hardwood, (**C**): Softwood).

**Figure 2 polymers-16-01750-f002:**
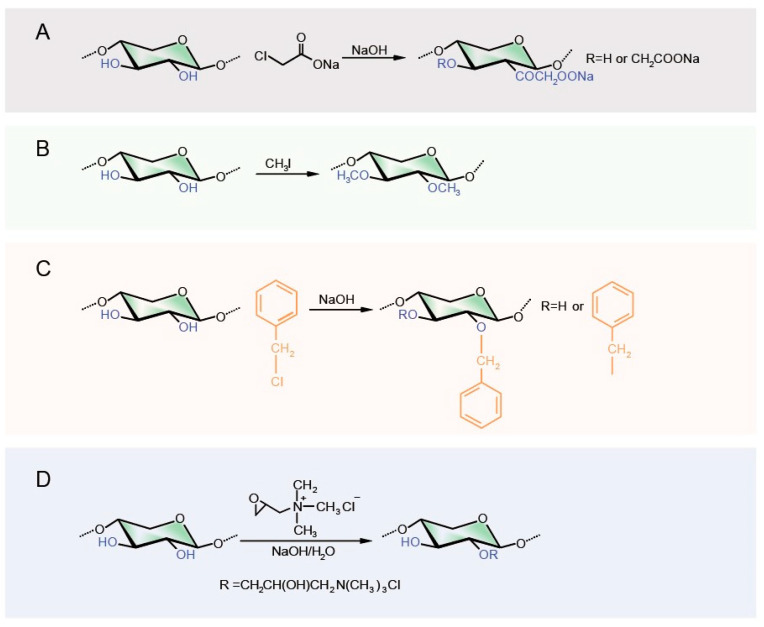
The etherification of hemicellulose ((**A**): Carboxymethylation, (**B**): Methylation, (**C**): Benzylation, (**D**): Quaternization).

**Figure 3 polymers-16-01750-f003:**
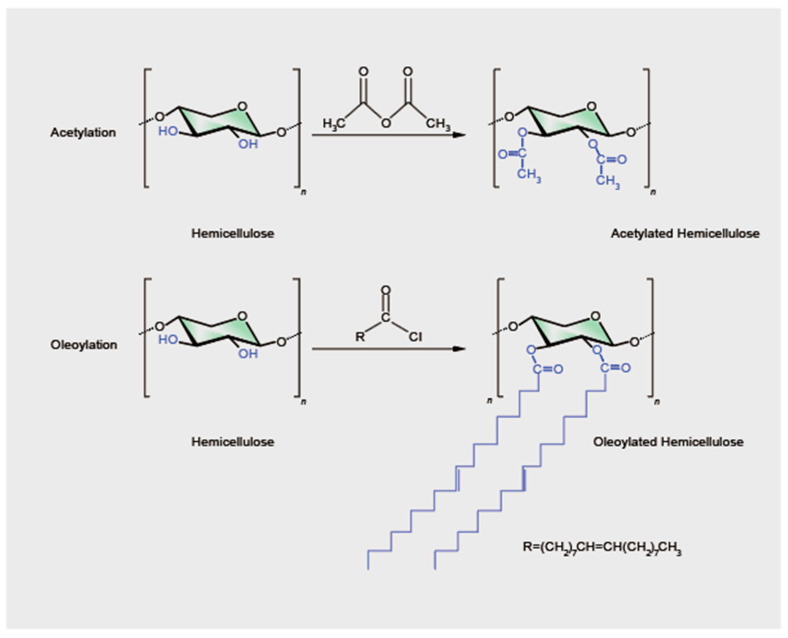
The esterification of hemicellulose.

**Figure 4 polymers-16-01750-f004:**
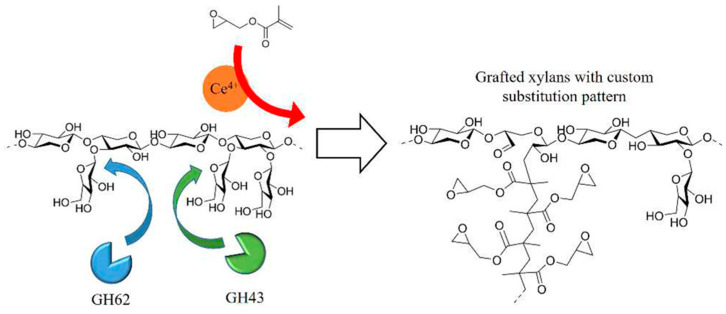
Using two α-arabinofuranosidases to remove α-arabinofuranose from wheat xylan [76]. Copyright 2024. Reproduced with permission from the American Chemical Society.

**Figure 5 polymers-16-01750-f005:**
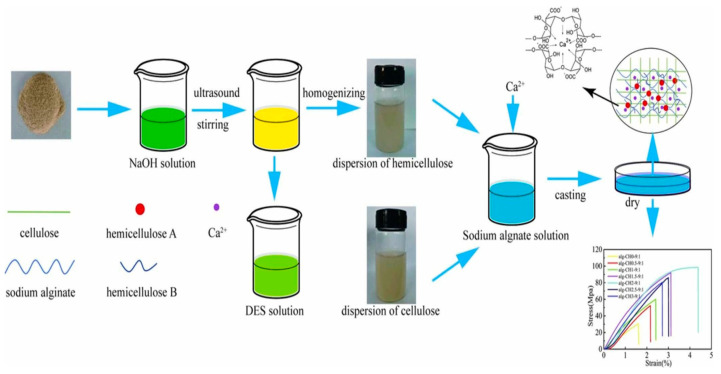
The synergistic enhancement of sodium alginate composite films by corn husk cellulose and hemicellulose [89]. Copyright 2024. Reproduced with permission from the Elsevier.

**Figure 6 polymers-16-01750-f006:**
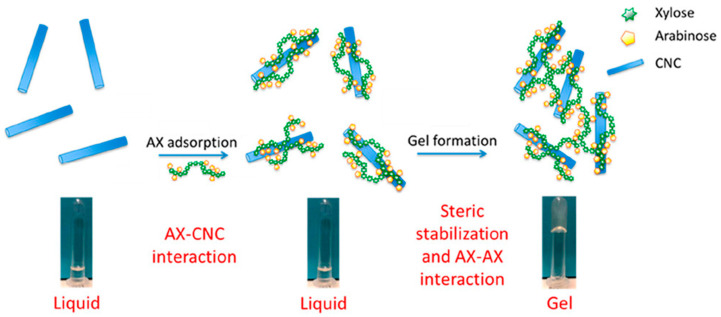
Construction of fully bio-based hydrogels using cellulose nanocrystals and arabinoxylan (CNC: cellulose nanocrystals, AX: arabinoxylan) [104]. Copyright 2024. Reproduced with permission from the American Chemical Society.

**Figure 7 polymers-16-01750-f007:**
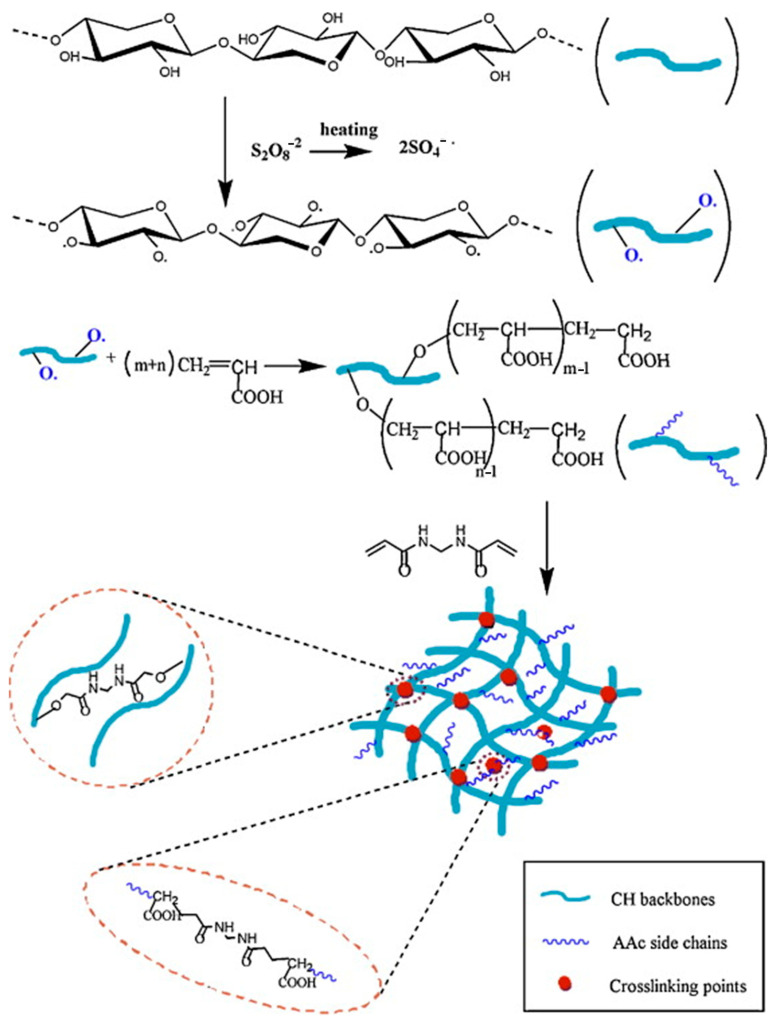
Synthesis of hemicellulose-co-PAAc hydrogels via radical copolymerization [105]. Copyright 2024. Reproduced with permission from the Elsevier.

**Figure 8 polymers-16-01750-f008:**
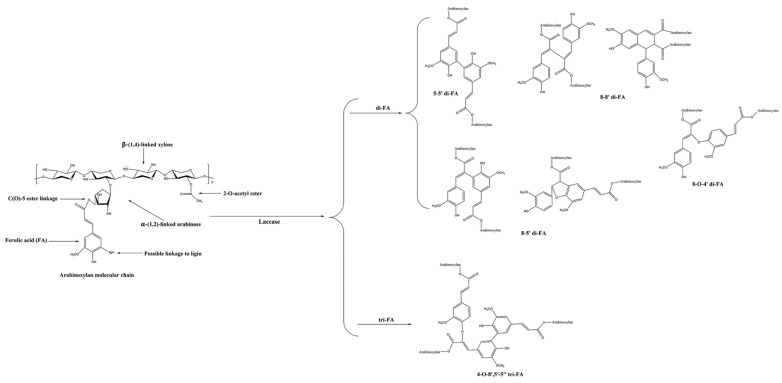
Laccase-induced cross-linking of xylan [51]. Copyright 2024. Reproduced with permission from the Elsevier.

**Table 1 polymers-16-01750-t001:** Yield range of hemicellulose from different lignocellulosic sources.

Lignocellulosic Sources	Hemicellulose Yield Range (Dry Weight)	Reference
Grasses	20–35%	[5]
Hardwoods	25–35%	[24]
Softwoods	18–35%	[25]
Straw	23–36%	[26]
Bamboo	24–34%	[25]
Coconut shells	24–28%	[27]
Wood chips and sawdust	15–35%	[28]

**Table 2 polymers-16-01750-t002:** The main limitations of hemicellulose from GOBPs.

Limitation	Description
Composition Complexity	Varied GOBP composition affects modification efficiency and quality. Impurities hinder optimal results.
Multi-Step Processes	Requires time-consuming, multi-step chemical reactions or complex physical treatments with specific equipment.
Property Deficiencies	Enhancements may cause deficiencies, e.g., acetylation improves meltability but reduces stability and antioxidant activity.
High Costs	Expensive reagents and equipment increase production costs, limiting large-scale applications.
Environmental and Energy Impact	Generates harmful by-products, high energy consumption due to required high temperatures and pressure.

## Data Availability

Not applicable.

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
