# Peer review of "Valorization of Grain and Oil By-Products with Special Focus on Hemicellulose Modification"

_polymers, 2024, doi:10.3390/polym16121750_

Round 1

Reviewer 1 Report

Comments and Suggestions for Authors

Dear Authors

Your article is a comprehensive review of grain and oil by-products - with special focus on hemicellulose modification.

The authors provide  well-organized Introduction to GOBP and its processing as well as their applications. It is a very well structured and comprehensive review provided a lot of information collected around from many references/authors.

In addition, the authors are aware about the limitations of hemicellulose materials that is also discussed in the article.

The conclusion part is adequately established based on the discussed topic.

Figures and Tables should be appropriately distributed t the bodytext of the manuscript.

Some of the Figures are small in size so it is difficult to be properly read/seen. Please, correct them in a suitable manner.

Some suggestions are given in the attached version of the article, that need clarification.

Thank you.

Reviewer 2 Report

Comments and Suggestions for Authors

The review “Valorization of grain and oil by-products: with special focus on hemicellulose modification” is devoted to one of the three polymers that make up lignocellulose mass, specifically hemicellulose. It should be noted that a review of the modification of hemicelluloses is a rather rare phenomenon. This manuscript seems to limit the diversity of plant materials to specific species, which the authors combine under the name “grain and oil by-products” = GOBP. The short review (only 25 pages) is a well-formatted analysis of 122 literature sources with illustrations. The relevance of the presented material is beyond doubt; readers are interested in the appearance of such high-quality reviews. But for publication it is necessary to correct several provisions that are presented in the list.

List of recommendations.

1. Introduction. The authors should list the names of lignocellulosic sources of hemicellulose, clearly indicate that the review will also consider wood sources of raw materials, and also limit or, conversely, expand this list to terrestrial plants and algae. It is recommended to provide general information on the yield of hemicelluloses in terms of raw materials.

2. Section 2. 1 Chemical modification of GOBP. It is recommended to supplement the requirement for the chemical purity of hemicelluloses for successful chemical modification.

3. Section 2. 2. 2 Ultrasonic. It is recommended to proofread this short text; there is either a typo (bracket) or the text is incomplete.

4. Section 2. 2. 3 Physicochemical modifications provides examples of enzymatic hydrolysis. It is recommended to indicate this directly in the title.

5. Section 4. Limitations of Hemicellulose Modification from GOBP is very valuable. It is recommended to structure these restrictions in the form of a short table.

6. Taking into account the recommendations made, the authors can strengthen the abstract by adding a sentence stating new scientific knowledge presented in the review.
